# Systemic and Mucosal Humoral Immune Responses to Lumazine Synthase 60-mer Nanoparticle SARS-CoV-2 Vaccines

**DOI:** 10.3390/vaccines13080780

**Published:** 2025-07-23

**Authors:** Cheng Cheng, Jeffrey C. Boyington, Edward K. Sarfo, Cuiping Liu, Danealle K. Parchment, Andrea Biju, Angela R. Corrigan, Lingshu Wang, Wei Shi, Yi Zhang, Yaroslav Tsybovsky, Tyler Stephens, Adam S. Olia, Audrey S. Carson, Syed M. Moin, Eun Sung Yang, Baoshan Zhang, Wing-Pui Kong, Peter D. Kwong, John R. Mascola, Theodore C. Pierson

**Affiliations:** 1Vaccine Research Center, National Institute of Allergy and Infectious Diseases, National Institutes of Health, Bethesda, MD 20892, USA; chcheng@nih.gov (C.C.); jboyingt2@aol.com (J.C.B.); edward.sarfo@icahn.mssm.edu (E.K.S.); cuiping.liu@nih.gov (C.L.); dkparchment@gmail.com (D.K.P.); andreabiju2020@gmail.com (A.B.); angelacorrigan0@gmail.com (A.R.C.); wangling@niaid.nih.gov (L.W.); shiw@mail.nih.gov (W.S.); yi.zhang3@nih.gov (Y.Z.); adam.olia@nih.gov (A.S.O.); audrey.carson@nih.gov (A.S.C.); syed.moin@nih.gov (S.M.M.); eunsung.yang@nih.gov (E.S.Y.); baoshan.zhang@nih.gov (B.Z.); wkong@mail.nih.gov (W.-P.K.); pdk3@cumc.columbia.edu (P.D.K.); john.mascola@modextx.com (J.R.M.); 2Icahn School of Medicine at Mount Sinai, New York, NY 10029, USA; 3Vaccine Research Center Electron Microscopy Unit, Cancer Research Technology Program, Frederick National Laboratory for Cancer Research Sponsored by the National Cancer Institute, Frederick, MD 21702, USA; yaroslav.tsybovsky@nih.gov (Y.T.); tyler.stephens@nih.gov (T.S.)

**Keywords:** vaccine, nanoparticle, SARS-CoV-2, lumazine synthase, mucosal immunity

## Abstract

**Background:** Vaccines that stimulate systemic and mucosal immunity to a level required to prevent SARS-CoV-2 infection and transmission are an unmet need. Highly protective hepatitis B and human papillomavirus nanoparticle vaccines highlight the potential of multivalent nanoparticle vaccine platforms to provide enhanced immunity. Here, we report the construction and characterization of self-assembling 60-subunit icosahedral nanoparticle SARS-CoV-2 vaccines using the bacterial enzyme lumazine synthase (LuS). **Methods and Results:** Nanoparticles displaying prefusion-stabilized SARS-CoV-2 spike ectodomains fused to the surface-exposed amino terminus of LuS were designed using structure-guided approaches. Negative stain-electron microscopy studies of purified nanoparticles were consistent with self assembly into 60-mer nanoparticles displaying 20 spike trimers. After two intramuscular doses, these purified spike-LuS nanoparticles elicited significantly higher SARS-CoV-2 neutralizing activity than spike trimers in vaccinated mice. Furthermore, intramuscular DNA priming and intranasal boosting with a SARS-CoV-2 LuS nanoparticle vaccine stimulated mucosal IgA responses. **Conclusion:** These data identify LuS nanoparticles as highly immunogenic SARS-CoV-2 vaccine candidates and support the further development of this platform against SARS-CoV-2 and its emerging variants.

## 1. Introduction

Since the outbreak of the coronavirus disease 2019 (COVID-19) pandemic, severe acute respiratory syndrome coronavirus 2 virus (SARS-CoV-2) has infected more than 750 million people worldwide and has caused over seven million deaths [1]. The continued transmission of SARS-CoV-2 and the evolution of immune-escape variants highlight a requirement for second-generation vaccines to achieve robust, broadly protective immunity that prevents virus acquisition [2]. The SARS-CoV-2 spike (S) is a metastable type I membrane protein responsible for host cell binding through interactions with the angiotensin-converting enzyme (ACE2), viral entry, and host-membrane fusion [3]. Previous research with the closely related Middle East respiratory syndrome coronavirus demonstrated that the prefusion conformation of the spike protein can be stabilized through the addition of 2 proline substitutions to enhance the elicitation of neutralizing antibody responses when used as vaccine immunogens. The stabilization of the SARS-CoV-2 spike with 2 or 6 proline substitutions (S2-P or S6-P, respectively) is similarly linked to protective antibody responses [4,5,6,7,8]. Multiple COVID-19 vaccine platforms have been developed to elicit spike-directed responses, including FDA-approved mRNA vaccines expressing the SARS-CoV-2 S-2P [9,10,11], mRNAs expressing a spike subunit [12], a DNA vaccine encoding a spike glycoprotein [13], live-attenuated [14] and inactivated viral vaccines [15,16], recombinant adenovirus and vesicular stomatitis platforms [17,18,19,20,21,22,23,24,25], a ferritin nanoparticle displaying spike glycoprotein [26], and recombinant spike proteins [27]. COVID-19 vaccines have proven highly effective at preventing severe disease and death [28,29]. However, their inability to prevent infection and transmission and the continued evolution of viral variants that limit vaccine or natural protection highlight a need for more effective second-generation, broadly protective vaccines.

The ordered array of multiple copies of an antigen on the surface of nanoparticles and viral vaccines can enhance immunogenicity and efficacy by several mechanisms [30,31]. The success of virus-like particle vaccines for hepatitis B and human papillomaviruses is attributed to this characteristic [32]. Compared to monovalent antigens, multivalent arrays of antigens may efficiently accumulate in lymph nodes, be efficiently taken up by antigen-presenting cells for cross-presentation, and support more robust activation of B and T cells [33,34,35,36,37]. Considerable effort has gone into the development of self-assembling, protein-based nanoparticle vaccine immunogens produced using genetic fusions [35,38,39,40,41,42,43,44,45,46] or the chemical conjugation of antigens to viral capsids, large multimeric enzymes, and de novo engineered multicomponent scaffolds [45,47,48]. Lumazine synthase (LuS) is an essential enzyme involved in riboflavin (vitamin B_2_) synthesis in microorganisms without a human homolog. LuS from the hyperthermophilic bacterium *Aquifix aeolicus* self-assembles from 60 subunits, each containing 154 amino acids, to form thermostable nanoparticles of 16 nm in diameter [49]. Both the amino- and carboxy-terminus of the LuS protein are exposed on the nanoparticle surface, allowing the genetic fusion and display of vaccine antigens. The proximity of the LuS N-termini to each of the 20 three-fold axes of the nanoparticle allows for the presentation of protein trimers. Preclinical and clinical studies have shown an HIV-1 vaccine candidate using the LuS platform elicits on-target immune responses [50]. The main rationales for choosing LuS 60-mer as the nanoparticle platform in our study are: (1) the bacterial enzyme has no human analogues with low risk for auto-immune reactions, (2) the 60-mer nanoparticles have already been safely tested in human phase I HIV-1 vaccine trials without severe adverse effect and (3) the 60-mer nanoparticle used here is thermostable, which is advantageous for manufacture and storage compared to other platforms.

Here, we constructed SARS-CoV-2 vaccines using the LuS 60-mer nanoparticle platform and characterized their antigenicity and immunogenicity. The expression of S-2P or S-6P LuS fusions in mammalian cells resulted in nanoparticles with stabilized pre-fusion spike trimers displayed on the surface capable of stimulating potent immune responses in mice. These data identify LuS nanoparticles as highly immunogenic SARS-CoV-2 vaccines and support the further development of this platform as vaccines against SARS-CoV-2 and its emerging variants.

## 2. Materials and Methods

### 2.1. Modeling of SARS-CoV-2 Spike and LuS Fusion

The cryo-EM structure of the S-2P trimer (PDB ID 6VXX) and the crystal structure of the LuS 60-mer (PDB ID 1HQK) were used as templates for vaccine design. The Cα positions of SARS-CoV-2 S C-terminal residue (1147) of each protomer of the trimer were superimposed onto the Cα positions of the LuS amino acid residue 1 on the surface of the icosahedral particle using LSQMAN [51]. The spike and LuS trimers were then moved apart along the trimer axis until there were no sidechain clashes between the SARS-CoV-2 spike and LuS. Finally, the program LOOPY was used to model the connections between each trimer [52]. For modeling and creation of molecular figures, the program UCSF CHIMERA was used [53].

### 2.2. Construction and Purification of LuS 60-mer Nanoparticles, S-2P Trimer Immunogens, and Antigens for ELISA Assay

Genes encoding a fusion between SARS-CoV-2 S protein and LuS (UniProt O66529) proteins using glycine-rich linkers of various lengths were codon optimized for expression in human cells and synthesized for cloning into expression vector pVRC8400 under the control of a CMV promoter (GenScript, Piscataway, NJ, USA). The *S* gene used in these fusions included residues 1-1147, 1-1206, or 1-308 (NTD domain only) of SARS-CoV-2 S (GenBank: MN908947) with proline substitutions at residues 986 and 987 and a “GSAS” substitution at the furin cleavage site (residues 682-685). Residues 44-47 in the LuS were mutated from REED to SGG to alleviate a potential clash with the glycine linkers connecting SARS-CoV-2 S and LuS. To express soluble stabilized S-2P SARS-CoV-2 S trimers, a construct encoding the SARS-CoV-2 protein detailed above was modified by the addition of a C-terminal T4 fibritin trimerization motif, an HRV3C protease cleavage site, a TwinStrepTag, and an 8× HisTag. A plasmid encoding residues 1-615 of human ACE2 fused with human Fc was generated to express human ACE2. Fc-tagged human ACE2, His-tagged SARS-CoV-2 RBD, and NTD recombinant proteins were expressed and purified using rProtA Sepharose and Ni-NTA resin (Cytiva, Marlborough, MA, USA), respectively, before sizing column purification [54,55].

Proteins and nanoparticles were produced in transiently transfected Expi293 cells (Thermo Fisher, Waltham, MA, USA) using Expi293fectin (Thermo Fisher, Waltham, MA, USA). For 1 L of Expi293 cell culture at a cell density of 2 million cells/mL, 1000 μg of expression plasmid was diluted in 50 mL Opti-MEM medium (Thermo Fisher, Waltham, MA, USA). A 3 mL volume of Expi293fectin (Thermo Fisher, Waltham, MA, USA) transfection reagent was diluted into 50 mL Opti-MEM medium and incubated for 5 min. Diluted transfection reagent was then added to diluted DNA (1000 μg) in a total volume of 50 mL and incubated for 15 min. The transfection reagent and DNA complex were added to 800 mL of Expi293 cells. One day later, 80 mL HyClone SFM4HEK293 medium (Cytiva, Marlborough, MA, USA) was added to each flask of 1 L cell cultures. Culture flasks were kept in a shaker incubator at 120 rpm, 37 °C, 9% CO_2_ for an additional 6 days. Protein was purified from filtered cell supernatants using *Galanthus nivalis* (GNA)-lectin gel (EY Laboratories, Inc., San Mateo, CA, USA) for LuS nanoparticles, or Strep-Tactin resin (IBA, Gottinggen, Germany) for spike trimer, or Protein A resin for human ACE2-Fc, before being subjected to additional purification by size-exclusion chromatography using a Superdex 200 10/300 Increase column (GE Healthcare, Chicago, IL, USA) in 2 mM Tris pH 8.0, 200 mM NaCl and 0.02% NaN_3_. The soluble S-2P trimers were incubated with 10% (*wt*/*wt*) HRV3C protease for 2 h at room temperature. Trimers were then passed over NiNTA resin to remove the cleaved tags and His-tagged protease before being run over a Superdex 200 10/300 Increase column (Cytiva, Marlborough, MA, USA) in 2 mM Tris pH 8.0, 200 mM NaCl, and 0.02% NaN_3_.

### 2.3. Animal Studies

Six- to eight-week-old BALB/c female mice were obtained from Charles River Laboratory (Wilmington, MA, USA) and housed and cared for in an American Association for Accreditation of Laboratory Animal Care-accredited facility at the VRC, NIAID, NIH in accordance with local, state, federal, and institute policies. For protein immunization studies, 50 μL of immunogen mix, containing a specified mass (μg) of filter-sterilized protein immunogen was mixed with 50 μL of Sigma adjuvant system (Sigma-Aldrich, St Louis, MO, USA, one vial dissolved in 1 mL PBS) and injected to the inner thigh of the two rear legs of each mouse. For DNA immunizations, 10, 2, or 0.4 μg of DNA plasmid in 100 μL PBS was split and administered intramuscularly to the two hind legs of anesthetized animals, followed by electroporation with the AgilePulse System (Harvard Apparatus, Holliston, MA, USA) using the manufacturer-recommended settings. The electroporation was applied to the muscles at the injection sites in both hind legs. Intranasal protein immunizations were performed with 2 μg of protein in a total of 20 μL solution with 10 μL applied to each nostril of an isoflurane anesthetized mouse using a 20 μL pipette tip. Serological studies were performed using blood collected weeks after each injection, or from mucosal surfaces following nasal wash or bronchoalveolar lavage (BAL). BAL was performed by repeated flushing and aspiration with PBS into the lungs of sacrificed animals. Nasal washes were collected by cannulation of the trachea of sacrificed mice. The nasal cavity of sacrificed mice was flushed three times with 0.5 mL of 1% BSA containing PBS (BSA–PBS, pH 7.4).

### 2.4. Antigenic Characterization and Serum Binding ELISA

Antigenic characterization of nanoparticle vaccines and S-2P trimers was performed by enzyme-linked immunosorbent assay (ELISA). For stability analysis, S-2P or nanoparticles were diluted to 4 µg/mL in PBS and incubated at different temperatures from 4 °C to 65 °C for 10 min. Their recognition by ACE2 and the monoclonal antibody of LY-CoV555 post the heat treatment was measured with ELISA. Costar-half area plates (Corning, Kennebunk, ME, USA) were coated with 50 µL/well of 50 ng S-2P LuS02, NTD-60-mer or S-2P and other indicated antigens in 1× PBS overnight at 4 °C. Between each subsequent step, unless otherwise stated, plates were washed five times with 150 µL/well of PBS-T (1× PBS plus 0.05% Tween). Plates were blocked with 100 µL/well of 5% skim milk + 2% BSA in 1× PBS for 1 h at room temperature. Serial dilutions of ACE2-hFc or the monoclonal antibodies (mAbs) 4-8 or 2-4 [56], LY-CoV555 [57], S309 [58] or H4 [59] were prepared at a starting concentration of 2 μg/mL in dilution solution (5% skim milk + 2% BSA in 1× PBS), and diluted in a 7-point, 5-fold scheme. A volume of 50 µL/well was added to plates and incubated for 1 h at room temperature. A volume of 50 µL/well of a 1:5000 dilution of goat anti-human IgG-HRP (Jackson Immunoresearch, West Grove, PA, USA) was added to plates and incubated at room temperature for 1 h. Plates were developed for 10 min at room temperature with 50 µL/well tetramethylbenzidine (TMB) substrate (SureBlue; KPL, Gaithersburg, MD, USA) and stopped with 50 µL/well 1N sulfuric acid (Fisher Chemical, Fair Lawn, NJ, USA), without washing. The absorbance of each well was measured at 450 nm (SpectraMax using SoftMax Pro, version 5, software; Molecular Devices, Sunnyvale, CA, USA). The measured optical densities (OD) were analyzed after subtracting non-specific goat anti-human IgG-HRP background activity. ELISAs to assess serum binding to S-2P and NTD or RBD subunit proteins were performed using a similar procedure with the following modifications. Plates were coated overnight at 4 °C with 50 ng of S-2P, NTD, or RBD in 1× PBS. After blocking for 1 h at room temperature, seven-fold serially diluted serum was added at 1:50 starting dilution. Each well was incubated with 50 µL/well of goat anti-mouse IgG secondary antibodies (Thermo Fisher, Waltham, MA, USA) at 1:5000, goat anti-mouse IgG2a secondary antibodies, or goat anti-mouse IgG1 antibodies for 1 h at room temperature before the binding was visualized chromatographically as detailed above. Endpoint titer was defined as the maximum serum dilution to maintain an OD reading of greater than 0.1.

### 2.5. Nasal and BAL Anti-S-2P IgG and IgA ELISA

Mucosal immune responses following an intramuscular (IM) or intranasal (IN) boost were assessed by measuring S-2P-binding antibodies present in nasal washes and bronchoalveolar lavage fluid (BAL) samples. Anti-S-2P IgG and IgA ELISAs were performed by coating 96-well MaxiSorp plates (Thermo Fisher Scientific, Waltham, MA, USA) overnight at 4 °C with 50 µL/well of 100 ng S-2P in 1× PBS. Between each subsequent step, plates were washed five times with 250 µL/well PBS-T (1× PBS plus 0.05% Tween) unless otherwise stated. Plates were blocked with 250 µL/well of 5% skim milk in 0.1% PBS-T for 1 h at room temperature. Next, nasal wash samples were diluted in dilution solution (2% skim milk + 0.1% PBS-T) at a 1:3 dilution ratio. A volume of 100 µL/well was added to the plates and incubated for 2 h at room temperature. BAL samples were diluted in dilution solution (1:6 dilution ratio), and 100 µL/well was added to the plates and incubated for 2 h at room temperature. A volume of 50 µL/well of goat anti-mouse IgG (H + L) HRP secondary antibody (Jackson Immunoresearch, West Grove, PA, USA) diluted 1:2000 or goat anti-mouse IgA Heavy Chain HRP secondary antibody (Novus Biologicals, Centennial, CO, USA) diluted 1:1000 in dilution solution was added to the plates. After 1 h incubation at room temperature, plates were washed three times with PBS-T. Next, plates were developed with 50 µL/well tetramethylbenzidine (TMB) substrate (SureBlue; KPL, Gaithersburg, MD, USA) for 15 min at room temperature, and the reaction was stopped with 50 µL/well 1 N sulfuric acid (Fisher Chemical, Fair Lawn, NJ, USA) without washing. Plates were read at 450 nm (SpectraMax using SoftMax Pro, Version 5, software; Molecular Devices, Sunnyvale, CA, USA), and the optical densities (OD) were analyzed following subtraction of the non-specific horseradish peroxidase background activity.

### 2.6. Neutralization Assay

Neutralization assays with spike-pseudotyped lentiviruses were performed as previously described [54]. To produce SARS-CoV-2 pseudoviruses, a codon-optimized CMV/R-SARS-CoV-2 S (Wuhan-1, Genbank #: MN908947.3) plasmid was constructed. Pseudoviruses were produced by co-transfection of plasmids encoding a luciferase reporter, lentivirus backbone, and *S* genes into HEK293T/17 cells (ATCC, Manassas, VA, USA, #CRL-11268). For SARS-CoV-2 pseudovirus, the human transmembrane protease serine 2 (TMPRSS2) plasmid was also co-transfected. Immune serum was mixed with pseudoviruses, incubated at 37 °C for 45 min, and added to ACE2-expressing 293T cells. Seventy-two hours later, cells were lysed, and luciferase activity was measured as relative light units (RLUs). Percent neutralization was normalized considering uninfected cells as 100% neutralization and cells infected with only pseudovirus as 0% neutralization. ID50 titers were determined using a log (agonist) versus normalized response (variable slope) nonlinear function in Prism v8 (GraphPad Software, San Diego, CA, USA).

### 2.7. Negative-Stain Electron Microscopy

Nanoparticles were diluted to a concentration of about 0.1 mg/mL with a buffer containing 10 mM HEPES, pH 7, and 150 mM NaCl. A 4.7 µL drop of the sample was applied on a glow-discharged carbon-coated copper grid for 15 s and removed with filter paper. The grid was washed three times by applying a drop of the same buffer, and adsorbed nanoparticles were negatively stained with 0.7% uranyl formate. Datasets were collected at a nominal magnification of 57,000× (pixel size: 0.25 nm) on a Thermo Scientific Talos F200C electron microscope (Waltham, MA, USA) operated at 200 kV and equipped with a Ceta camera (Thermo Fisher Scientific, Waltham, MA, USA). Particle picking was performed using Topaz [60]. Relion was used for reference-free 2D classification [61].

### 2.8. Statistical Analysis

All analysis was performed using GraphPad Prism v8.4.1 (GraphPad Software, Inc., Boston, MA, USA). Two-tailed Mann–Whitney tests were used to compare two experimental groups. A Spearman correlation test was performed to correlate neutralizing titers with ELISA endpoint titers. ns = *p*-value > 0.05, * = *p*-value < 0.05, ** = *p*-value < 0.01, *** = *p*-value < 0.001, **** = *p*-value < 0.0001.

## 3. Results

### 3.1. Design of SARS-CoV-2 Nanoparticle Immunogens Based on LuS

SARS-CoV-2 spike-LuS nanoparticle immunogens were designed using available structures of the soluble S-2P trimer (PDB ID 6VXX) and LuS 60-mer (PDB ID 1HQK). Constructs were designed to connect the S-protein ectodomain C-terminal region (residue 1147 or 1206) to the N-terminus (Met1) of LuS using three-, seven-, or fifteen-residue glycine-rich linkers (Table 1). Since neutralizing antibodies also target the amino-terminal domain (NTD) of the S protein [56,62,63,64], we also designed LuS nanoparticles linked to the NTD domain (residues 1-308) using 14-residue glycine-rich linkers (Figure 1 and Table 1).

### 3.2. Expression and Characterization of the SARS-CoV-2 LuS Nanoparticle Immunogens

Genes encoding SARS-CoV-2 LuS nanoparticle designs were human-codon optimized, cloned into a eukaryotic expression vector, and expressed in Expi293 cells. As there are 22 potential N-linked and two potential O-linked glycosylation sites per spike monomer, the spike glycoprotein contains many α-1,3 mannose residues, which can be bound by *Galanthus nivalis* lectin. Nanoparticles were affinity-purified from transfection supernatants using lectin resins to selectively bind glycosylated S proteins and subjected to size exclusion chromatography (SEC) to separate the nanoparticles from smaller contaminants. The purification yield is detailed in Table 1. Further optimization of purification methods through multiple elutions of bound nanoparticles on the *Galanthus nivalis* lectin resins increased the recovery of S-2P-LuS02 (with a 7-amino-acid linker) and NTD-LuS, but not S-2P-LuS01 (with a 3-amino-acid linker). To improve yield, we next constructed nanoparticles that display the S ectodomain with additional stabilizing substitutions. Prior studies demonstrate the addition of four additional prolines (F817P, A892P, A899P, A942P) more efficiently stabilizes the S protein in the pre-fusion conformation [6]. Two constructs were designed that attach the S-6P protein to the N-terminus of LuS using 7- or 15-residue glycine-rich linkers (S-6P-LuS13 and 14). Production and purification studies revealed a yield of 3.44 and 2.03 mg/L for these new nanoparticles, respectively.

We next probed the structural integrity of the nanoparticle vaccines using negative stain electron microscopy (NS-EM). Images from 2D class averages of purified S-2P and S-6P LuS nanoparticles revealed ~50 nm nanoparticles with spike trimers displayed symmetrically on the surface of the 16 nm LuS core (Appendix A). The size and shape of the S trimers were consistent with a pre-fusion conformation.

The binding of S-2P nanoparticle vaccines to the SARS-CoV-2 receptor (ACE2) and human monoclonal antibodies (mAbs) was characterized with an ELISA (Figure 2A). S-2P-LuS02 bound ACE2 at a level similar to recombinant S-2P trimers. Efficient binding of this nanoparticle to all four anti-RBD mAbs (2-4, LY-CoV555, H4, and S309) and to the anti-NTD mAb 4-8 was observed. In each case, RBD mAbs bound S-2P on the nanoparticle more efficiently than an equal mass of the control S-2P trimer, perhaps due to avidity effects (Figure 2A). As expected, the NTD-LuS nanoparticle bound only to the NTD-specific mAbs 4-8. This mAb bound the NTD nanoparticle more efficiently than either the S-2P nanoparticle or the recombinant NTD monomer. Similar approaches were used to characterize the antigenic structure of S-6P nanoparticles. Wild-type Wuhan-1 S-6P nanoparticles bound efficiently to ACE2 and mAbs specific for the RBD or NTD. Nanoparticles displaying the beta B.1.351 variant S-6P trimer bound to ACE2 and S309 but did not bind to anti-NTD mAb 4-8 or Ly-CoV555 due to B.1.351 mutations in the NTD and RBD domains (Figure 2B). To test the thermostability of the vaccines, S-2P soluble trimers, S-2P-LuS02, and S-6P-LuS13 nanoparticles were incubated at temperatures ranging from 4 °C to 65 °C (Appendix A). All three vaccines exhibited strong binding to ACE2 after 10 min of exposure to 4 °C, 45 °C, and 50 °C; heating to 65 °C significantly reduced the binding. The S-6P-LuS13 nanoparticles maintained relatively better binding to ACE2 than S-2P and S-2P-LuS02, particularly after exposure to elevated temperatures of 45 and 50 °C. All three vaccines efficiently bound the LY-555 antibody after incubation at all temperatures tested, even at 65 °C.

### 3.3. Immunogenicity of SARS-CoV-2 LuS Nanoparticle Immunogens in Mice

To evaluate the immunogenicity of the SARS-CoV-2 LuS platform, we immunized BALB/c mice with two doses of 2, 0.4, or 0.08 μg of S-2P-LuS02 or S-6P-LuS14 nanoparticle vaccines at an interval of three weeks. Three additional groups of animals received the same mass of purified S-2P trimer as a control. All vaccines were adjuvanted with the Sigma Adjuvant System (SAS), formerly known as Ribi (Figure 3A). Serum S-2P protein-binding and neutralizing antibodies were evaluated using ELISA and lentivirus-based pseudovirus neutralization assays, respectively. Analysis of S-2P-binding antibodies in serum obtained two weeks after the first immunization revealed S-6P-LuS14-elicited detectable anti-S and anti-RBD responses even at low vaccine doses (0.08 and 0.4 μg) (Figure 3B,C). In contrast, S-2P-LuS02 nanoparticles and S-2P trimers did not elicit any RBD-directed antibody responses post the first dose. After the second boost, anti-S and anti-RBD responses increased in all groups. Notably, the nanoparticles (S-2P-LuS02 or S6-P-LuS14) generated significantly higher anti-RBD IgG responses as compared to the S-2P trimer at the lowest dose (Figure 3C right panel). Neutralizing activity in the immune sera was measured using a Wuhan1 strain pseudotyped virus at two weeks post-second immunization. The two nanoparticle antigens elicited potent neutralizing activity at all doses. At the lowest vaccine dose (0.08 μg), nanoparticle immunogens elicited significantly (*p* < 0.01) higher neutralizing activity (geometric mean ID_80_ titer of 305 and 204 for the S-2P nanoparticles and S-6P nanoparticles, respectively) than the recombinant trimer, which did not elicit detectable neutralizing activity (<40, Figure 3D). The nanoparticles are at least 5-fold more potent than S-2P trimers as measured with ID_80_ titers.

### 3.4. SARS-CoV 2-LuS Variants Elicited Enhanced Anti-IgG and Potent Neutralizing Responses

We next explored the impact of heterologous S nanoparticle boosting on the magnitude and breadth of vaccine-elicited antibody responses. Groups of mice were first primed with 1 μg of nanoparticles displaying NTD (NTD-LuS) or two proline-stabilized S-protein nanoparticles (S-2P-LuS01 or LuS02). Immunization with recombinant S-2P timers was included as a control. All regimens were formulated with SAS as an adjuvant (Figure 4A). Three weeks later, all groups except those receiving a primary S-2P LuS01 immunization received a second dose of the same vaccine. Vaccine-elicited antibody responses were measured by ELISA and lentivirus-based neutralization assay five weeks after the first dose (Figure 4). Anti-S and anti-RBD IgG responses were significantly higher for the S-2P-LuS02 and recombinant S-2P trimer groups as compared to animals receiving NTD-LuS. The NTD-LuS nanoparticle was as immunogenic as S-2P-LuS in eliciting anti-NTD IgG responses (Figure 4B). Both S-2P-LuS01 and S-2P-LuS02, with a three- or seven-amino acid linker, elicited a similar level of anti-S-2P IgG responses post one immunization (Figure 4D). All groups generated balanced Th1 and Th2 responses, as measured with the ratio of IgG2a/IgG1 (Appendix A). Serum neutralizing activity was similar post two immunizations for the S-2P-LuS02 or S-2P trimer immunized group (Figure 4C). Remarkably, the S-2P-LuS02 group elicited cross-neutralizing responses to SARS-CoV-1 pseudoviruses by Week 5 in three animals (Appendix A). There was a positive correlation between serum anti-SARS-CoV-1 S IgG titers and serum neutralization ID_50_ titers against SARS-CoV-1 pseudovirus (Appendix A).

All groups of mice were next boosted with 1 μg of S-6P-LuS B.1.351 nanoparticles 38 weeks after the first immunization, resulting in a significant increase (25- to 126-fold) in S-protein-binding titers (Figure 4D,E). Importantly, these levels surpassed the peak reached after the initial two immunizations at Week 5. Analysis of neutralization activity also revealed a marked increase in titer when assayed using the Wuhan strain, matching the S proteins included in the first dose, although titers in mice that received the NTD priming immunogen were very low (Figure 4F). Similarly, serum from all the groups exhibited enhanced neutralizing activity against SARS-CoV-2 variants of concern (VOC), namely Alpha (B.1.1.7), Beta (B.1.351), and Delta (B.1.617.2) variants (Figure 4F) when compared to Week 31 titers. Although the S-2P trimer primed group had neutralizing activity, it was not statistically significantly different from titers in sera prior to the B.1.351 S-6P LuS boost due to animal variations. The NTD-LuS primed group again showed the lowest neutralizing activity to the VOC after the variant immunogen boost.

### 3.5. Immunogenicity of DNA Immunization

To test the immunogenicity of LuS nanoparticles when delivered using a nucleic acid platform, we produced DNA plasmids encoding full-length SARS-CoV-2 S protein (pS-2P), or the nanoparticle vaccines S-2P-LuS and S-6P-LuS. The pS-2P (full-length) DNA construct encodes the same amino acid sequence as the Moderna/mRNA1273 and Pfizer-BioNTech vaccines. Groups of mice were immunized with three different doses (10, 2, and 0.4 μg) of pS-2P, pS-2P-LuS, and pS-6P-LuS by intramuscular electroporation at Weeks 0 and 3 (Figure 5A). Serum was tested two weeks after each immunization for antibodies that bind to S-2P trimer or RBD proteins, or neutralize viral pseudotypes made with Wuhan-1 S proteins (Figure 5B). Vaccination elicited considerable anti-spike IgG titers in all groups. After the first immunization, at 10 μg, titers of S- and RBD-binding antibodies were significantly higher for nanoparticle groups than the pS-2P full-length control. At this same time point, the nanoparticle groups also elicited significantly higher anti-RBD IgG responses than the pS-2P full-length group when administered at 2 μg doses (Figure 5B). The pS-6P LuS group elicited significantly higher (2 to 10-fold) neutralizing activity against Wuhan-1 pseudotyped viruses as compared to the pS-2P full-length protein or the pS-2P LuS nanoparticle vaccine (Figure 5C). Furthermore, the anti-RBD IgG titers positively correlated with the neutralization titers (r = 0.76, *p* < 0.0001, Figure 5D). The 0.4 μg immunized groups were terminated due to low serum-neutralizing activities, and the rest of the groups were further boosted as detailed in the next section.

### 3.6. Intranasal (IN) Nanoparticle Boosting Elicits a Robust Mucosal Immune Response

To investigate if intranasal (IN) nanoparticle immunization boosts mucosal responses in previously vaccinated animals, we administered 2 μg IN S-2P or S6-P nanoparticle protein boosts to the 2 μg DNA-vaccine-dosed groups detailed in Figure 5. Animals that received 2 μg of DNA pS-2P were also boosted with 2 μg of IN S-2P trimer proteins (Figure 6A). For comparison, a third IM dose of 10 μg DNA was administered to animals primed with 10 μg DNA described above (Figure 6A). All boosts were performed 36 months after the first vaccine administration. Serum-neutralizing activity was similar for all groups after the boost, with the nanoparticle groups trending to higher titers (Figure 6B). Analysis of S-protein-binding IgG present in nasal washes or BAL revealed similar levels in all dose groups and both boosting regimens. In contrast, animals receiving an intranasal nanoparticle boost, but not intramuscular DNA, elicited anti-S-2P IgA responses. The highest responses were observed in the twice IM pS-6P LuS primed and IN nanoparticle-boosted animals (Figure 6C). IgA responses in the BAL were higher for S-2P-LuS and S-6P-LuS groups than animals boosted with recombinant trimers.

## 4. Discussion

In this study, we characterize the immunogenicity and versatility of LuS-based nanoparticle vaccines. Previously, nanoparticle platforms incorporating two or more protein components have been evaluated as SARS-CoV-2 vaccines [44,48]. These are produced by attaching immunogens to a protein nanoparticle core, as exemplified by the SpyTag-SpyCatcher conjugation system, or as de novo engineered multicomponent systems. Both approaches are promising but require producing each component in different cells, using different purification processes, followed by in vitro assembly and final purification of the nanoparticles. Virus-like particles (VLPs) in which full-length spikes are embedded within lipid membranes have also been constructed [27]. However, a complication of this platform is the presence of producer cell-derived membrane proteins incorporated into the VLPs. To limit manufacturing challenges, here, we constructed a single-component self-assembling nanoparticle by genetic fusion of the spike protein to a bacterial lumazine synthase nanoparticle platform. Since the native LuS is a cytoplasmic protein, we used the signal peptide from the spike protein to direct LuS into the secretory pathway of mammalian cells. This approach allows for delivery as a recombinant nanoparticle or as a nucleic acid vaccine. No human homologs for LuS exist, as mammals lack enzymes to synthesize vitamin B2, thus limiting the potential of off-target auto-immune responses in vaccinated humans. The LuS 60-mer platform has previously been used for an HIV-1 vaccine incorporating genetic fusion with the gp120-derived eOD-GT8 immunogen. The self-assembling eOD-GT8 60-mer has been GMP-manufactured and shown to be safe, tolerable, and immunogenic in eliciting VRC01 precursor B-cells in a recent Phase-1 trial [50]. Anti-LuS immune responses were detected in eOD-GT8 60-mer vaccinated participants without adverse effects [65]. As the hyperthermophilic bacterium *Aquifix aeolicus* is not a member of the human microbiome, pre-existing immunity to LuS 60-mer in humans and the risk of autoreactivity is low. Thus, the single-component LuS genetic fusion platform is a technology with the potential to be widely applicable to vaccines for other emerging and re-emerging pathogens.

Here, we constructed a small panel of nanoparticles displaying SARS-CoV-2 S protein as a vaccine immunogen. Nanoparticles were constructed initially using S proteins of the Wuhan strain stabilized using the two-proline approach incorporated into the Moderna and Pfizer vaccines [11], or with six proline substitutions [6,8]. Recombinant hexaproline (6P) trimers are more stable, support production at higher yields, and are more efficacious vaccines in mice and hamsters than current vaccines that incorporate diproline (2P) mutations [6,8,66]. In agreement with these findings, S-6P nanoparticles demonstrated superior antigenicity and immunogenicity compared to S-2P when presented on recombinant nanoparticles or delivered by DNA vaccines. S-6P nanoparticle vaccines elicited robust anti-RBD responses two weeks after immunization, which correlated positively with neutralization titer (Appendix A). As shown in Figure 4, serum-neutralizing activity peaked at Week 5 after the second immunization and decreased by 10-fold by Week 31. We did not collect data from a sufficient number of time points to estimate the durability of the response with precision. This limited durability of the response may be animal model-specific or modulated using an optimal adjuvant. Further studies are required to measure and potentially extend the durability of the S-protein antibody response. The immune responses elicited by our nanoparticle vaccines are similar to the responses by other viral-like particles, including the tobacco mosaic virus-like nanoparticle [67] and the murine leukemia virus (MLV)-based enveloped virus-like particles (eVLPs) [68]. A third boost with a B.1.351 variant nanoparticle vaccine significantly enhanced ELISA IgG titer and neutralizing activity to the Wuhan strain and multiple variants (alpha, beta, delta), suggesting that heterologous cross-variant protection is feasible. The nanoparticle vaccines can also be updated to include contemporary S-protein sequences.

Although NTD-targeted neutralizing antibodies have been observed in individuals infected with SARS-CoV-2, the NTD-LuS 60-mer vaccine failed to elicit any neutralizing activity in mice. This may be due to the immunodominance of non-neutralizing NTD epitopes in this model, or the inability of NTD 60-mer-specific antibodies to bind to native S trimers on SARS-CoV-2 pseudoviruses. Based on molecular modelling, it is estimated that 31% of the S-2P-LuS nanoparticle surface is covered with the spike, while for NTD-LuS nanoparticles, the surface is 35% covered by the NTD. Although the S-protein trimer is considerably larger than the NTD protein, the actual LuS surface area exposed in the S-protein nanoparticle versus the NTD nanoparticle is quite similar. This is in part because the S-protein monomers are held tightly together in their respective trimers, particularly at the trimer base, and also because of the flexibility provided by the glycine-rich linkers. Furthermore, S-2P nanoparticles elicited an NTD-binding antibody response similar in magnitude to the NTD 60-mer vaccine, suggesting there is no advantage of an NTD-focused immunogen.

FDA-approved SARS-CoV-2 vaccines provide outstanding protection against severe disease and hospitalization. However, they do not prevent infection or transmission and require booster vaccinations to enhance durability [2]. These FDA-approved vaccines are given IM and elicit relatively strong systemic antibody responses but trigger limited mucosal IgA responses. A locally generated mucosal immune response in the respiratory tract may more efficiently prevent acquisition and, therefore, have a greater epidemiologic impact on the course of the global epidemic. Aerosol and IN immunization elicit both mucosal and systemic immunity against respiratory pathogens, including SARS-CoV-2 [69,70], with two vaccines based on recombinant adenoviral vectors approved in China [71] and India [72]. Prior studies in rodent models demonstrate that IN immunizations can be superior to IM immunizations, as reflected by both systemic and mucosal responses, and improved protection against viral challenge [73,74,75,76]. Furthermore, recent studies in a non-human primate model establish that significant and durable protection from heterologous SARS-CoV-2 acquisition can be achieved through aerosolized delivery of an S-encoding adenovirus vector, with mucosal IgA correlating strongly with protection in this model [77]. In our study, we demonstrate that S-6P nanoparticle boosting regimens elicited significant IgA titers in the nasal cavity and BAL in the absence of a mucosal adjuvant. The neutralizing activity of the mucosal samples was not determined in this study due to interference of the mucosal fluid in the neutralization assay. Future studies will explore optimal nanoparticle regimens to prevent SARS-CoV-2 infection in animal models and the neutralizing activity in the mucosal compartment. Furthermore, the exact level of IgA needed for protection and the durability of the IgA response are not known. A proper mucosal adjuvant should further boost such responses both for potency and longevity for protective effect. Compared to the reported ID_50_ neutralizing titer achieved with mRNA vaccine in mice (819, geometric mean), the S-6P-LuS14 nanoparticles elicited a similar range of high titers (ID_50_ geometric means from 487 to 1134, depending on vaccine dose; Figure 3D). Although we used transient transfection to produce these nanoparticle vaccines with a vaccine yield of above 1 mg/L, the single-component self-assembling characteristics of this platform readily support the creation of a stable producer cell line to support large-scale production. Similar large-scale manufacturing has been standardized for commercial antibody production, which is applicable. Although we have not measured the protective efficacy in an animal challenge experiment, it has been shown that a serum ID_50_ titer of 819 protects mice from SARS-CoV-2 challenge. The ID_50_ titer achieved in this study is within the protective range. Future studies will explore optimal nanoparticle regimens to prevent SARS-CoV-2 infection in animal models and the neutralizing activity in the mucosal compartment.

## 5. Conclusions

We constructed a versatile, single-component, self-assembling nanoparticle vaccine for SARS-CoV-2 using the LuS 60-mer nanoparticle platform. We demonstrate that stabilized S proteins displayed by LuS nanoparticles are highly immunogenic when delivered as recombinant purified protein or DNA vaccine via IM or IN routes. This promising vaccine platform complements ongoing efforts to develop second-generation vaccines and vaccination regimens to prevent SARS-CoV-2 acquisition and is applicable to other emerging and re-emerging pathogens.

## Figures and Tables

**Figure 1 vaccines-13-00780-f001:**
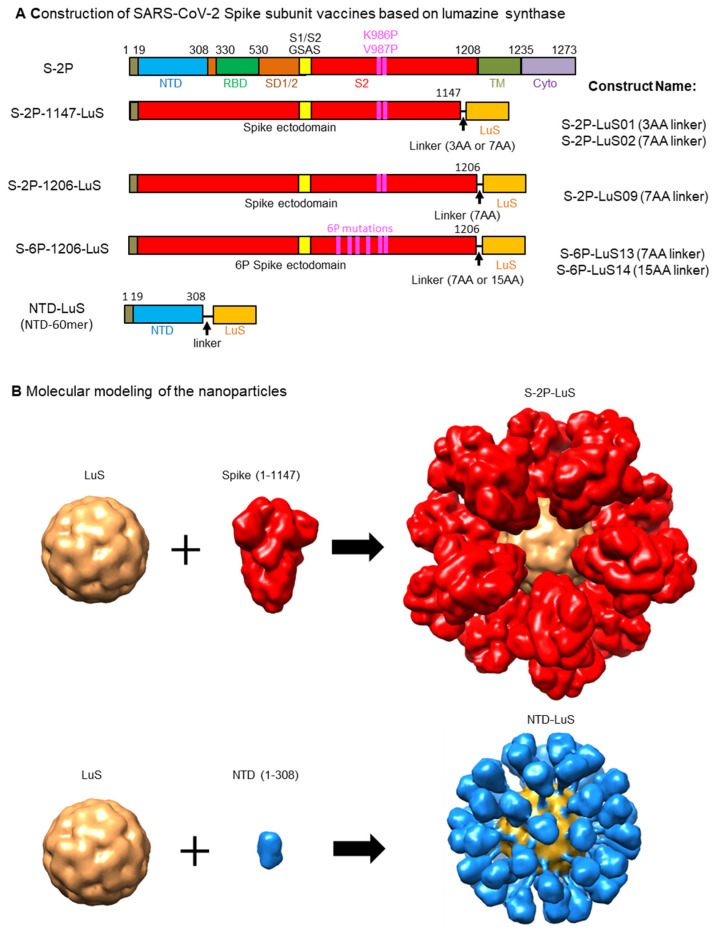
Design and molecular modeling of vaccine constructs. (**A**) Schematic representation of the S-2P full-length trimer. A “GSAS” substitution (yellow box) eliminates cleavage by furin between S1 and S2 domains (residues 682–685). Proline substitutions at residues 986 and 987 stabilize the S protein in the prefusion conformation. Amino acid (AA) positions for known S subunits are labelled on the top of the construct: amino acid 1-19: signal peptide; NTD: N-terminal domain; RBD: receptor binding domain; SD1/2: subdomains 1 and 2; S2: S2 domain; The full length of S also include TM: transmembrane domain; Cyto: cytoplasmic domain. The S-2P constructs contain two proline mutations at amino acid positions 986 and 987, while the S-6P constructs contain four additional proline mutations (F817P, A892P, A899P, A942P), as indicated by the purple boxes. The ecotodomain of S from amino acid 1-1147 or 1-1206 is fused to the N-terminus of lumazine synthase (LuS) in S-2P-1147-LuS or S-2P-1206-LuS, respectively. NTD-LuS contains the NTD and LuS fusion. (**B**) Molecular modeling of S-2P-LuS and NTD-LuS nanoparticles. Constructs were modeled using existing structures of the S-2P trimer (PDB ID 6VXX) and LuS 60-mer (PDB ID 1HQK). 3AA linker: GSG; 7AA linker: GSGGGSG; 15AA linker: GGSGGSGGSGGSGGG; Linker for NTD-LuS: GGSGGSGGSGGSGG.

**Figure 2 vaccines-13-00780-f002:**
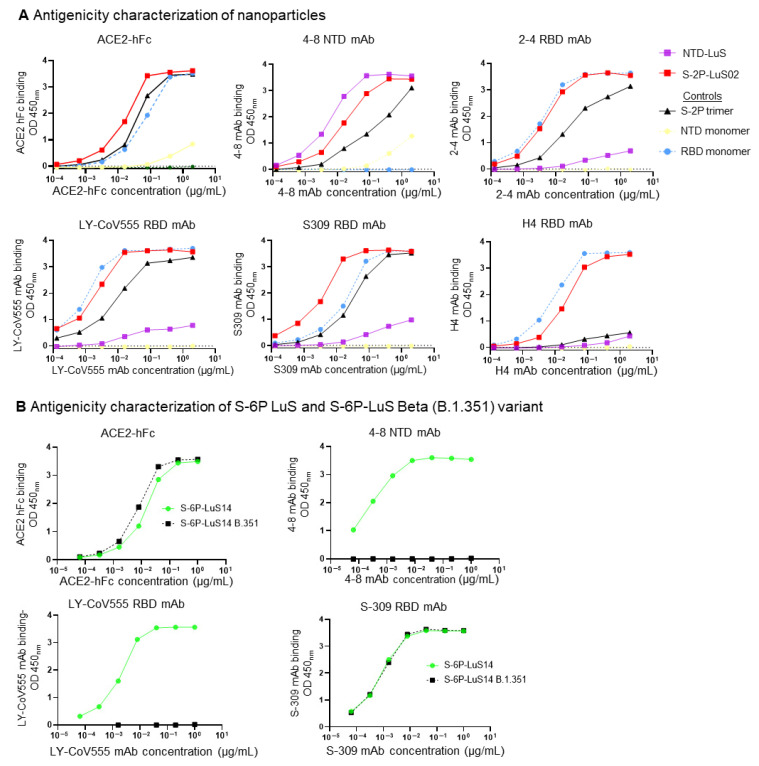
Antigenic characterization of nanoparticles. (**A**) Four RBD-specific antibodies (2-4, LY-CoV555, S309, and H4), one NTD antibody (4-8), and ACE2 fused with hFc were used to assess the antigenicity of these nanoparticles. S-2P trimers, RBD monomers, and NTD monomer proteins were included in these studies as control proteins. (**B**) Antigenic characterization of SARS-CoV-2 Beta variant B.1.351 and S-6P-LuS14 using two RBD mAbs (LY-CoV555, S-309), one NTD mAb (4-8), and ACE2 fused with hFc.

**Figure 3 vaccines-13-00780-f003:**
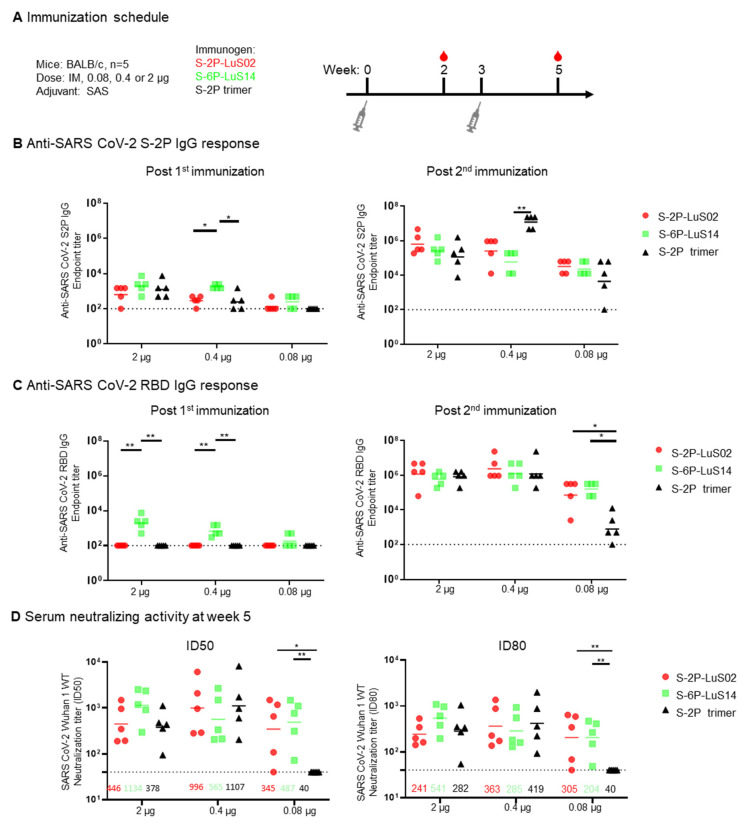
Immunogenicity of SARS-CoV-2 LuS Nanoparticle Immunogens in Mice. (**A**) Immunization scheme for groups of mice (n = 5 per group) immunized with different doses (0.08, 0.4 or 2 ug) of nanoparticle or recombinant trimer immunogens. (**B**) The S-2P protein binding or (**C**) RBD-binding IgG response as measured by ELISA as shown at Week 2 (**left panel**) and 5 (**right panel**). (**D**) Serum neutralization of a SARS-CoV-2 Wuhan strain pseudotype was measured with geometric means for neutralization titers shown for each group. Data (**B**–**D**) were analyzed by Kruskal–Wallis test followed by Dunn’s multiple comparison test. (* *p*  <  0.05; ** *p*  <  0.01). Dashed lines represent the lowest serum dilution at 1:40 in the neutralization assay as the detection limits.

**Figure 4 vaccines-13-00780-f004:**
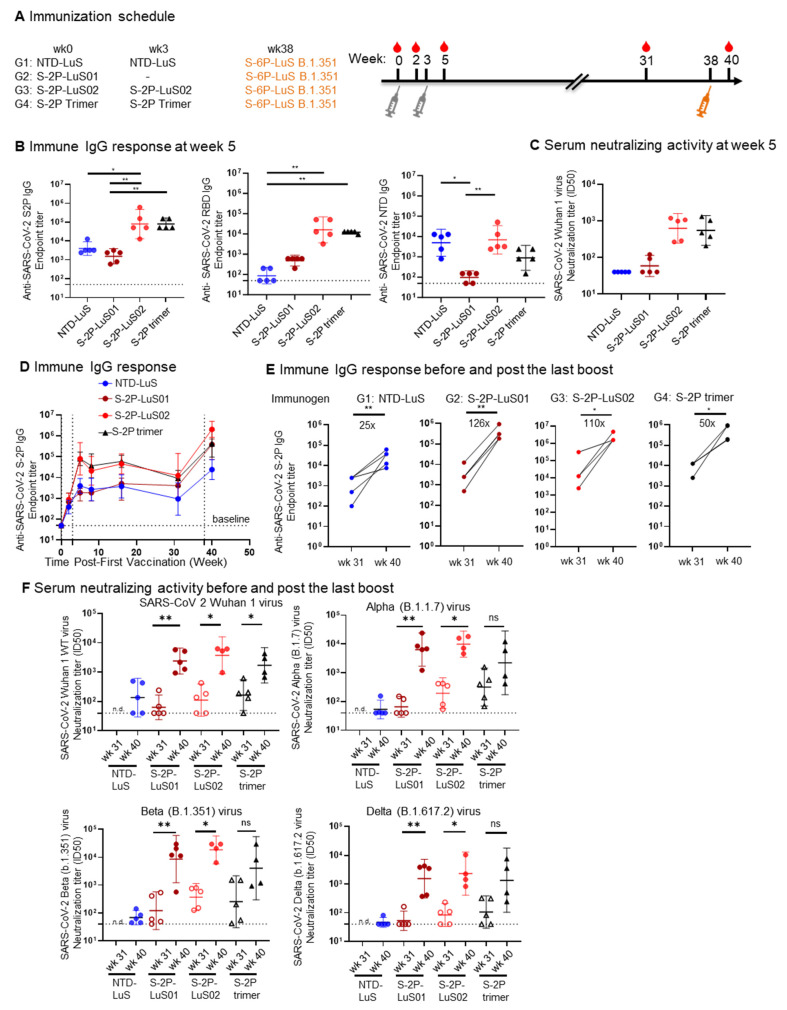
Potent immune responses against Wuhan WT and VOC were elicited in mice. (**A**) Immunization scheme for four groups of mice (n = 5) with 1ug of the indicated immunogens. (**B**) The elicited IgG response as measured by ELISA is shown at Week 5 (post 2nd immunization) to SARS-CoV-2 spike (**left panels**), RBD (**middle panel**), and NTD (**right panel**) antigens. (**C**) Week 5 serum neutralization activity (ID_50_) against Wuhan WT SARS-CoV-2 pseudovirus. (**D**) Serum anti-S-2P spike IgG titers time course from Weeks 0 to 40. (**E**) serum anti-SARS-CoV 2 S-2P IgG responses pre and post S-6P LuS B.1.351 boost. (**F**) Serum neutralization activity (ID_50_) against pseudotyped Wuhan WT SARS-CoV-2 and VoC (alpha, beta, delta) at pre (wk31) and post (wk 40) S-6P Beta Variant boost. Geometric mean and 95% CI were shown. Data for panel (**B**) were analyzed by Kruskal–Wallis test followed by Dunn’s multiple comparison test, data (**E**,**F**) were analyzed with two-tailed Mann–Whitney test. (* *p*  <  0.05; ** *p*  <  0.01).

**Figure 5 vaccines-13-00780-f005:**
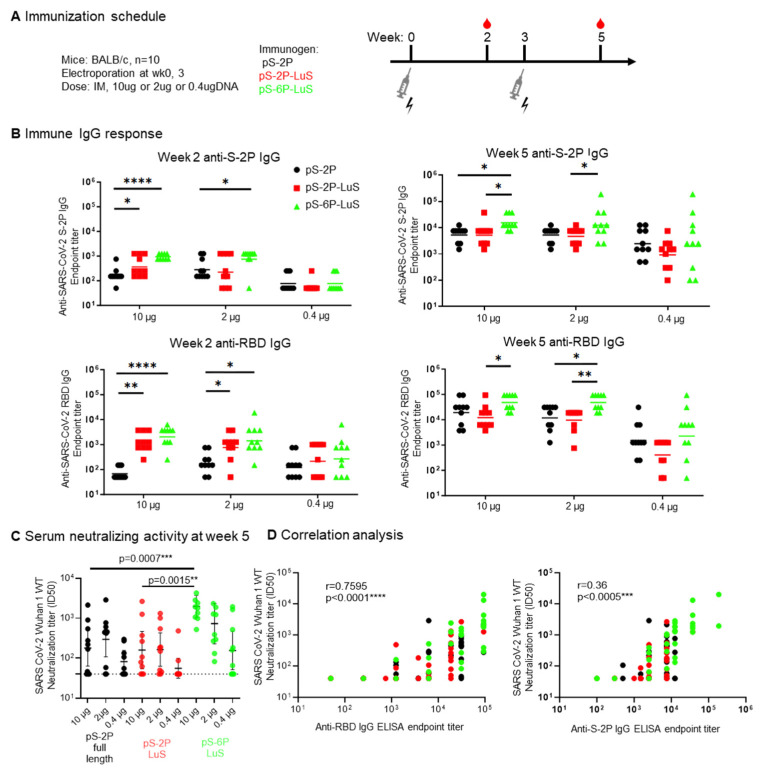
DNA delivery of S nanoparticle immunogens elicited immune responses. (**A**) Immunization scheme for nine groups of mice (n = 10) immunized with either 10, 2, or 0.4 µg of indicated DNA plasmids twice intramuscularly. (**B**) SARS-CoV-2-S2P and SARS-CoV-2-RBD IgG response assessed by ELISA at Week 2 or Week 5. (**C**) Week 5 serum neutralization and (**D**) two-sided Pearson correlation of anti-RBD (left panel) or anti-S-2P (right panel) IgG ELISA endpoint titer to SARS-CoV-2 Wuhan 1 WT neutralization titer. Data (**B**,**C**) were analyzed by Kruskal–Wallis test followed by Dunn’s multiple comparison test. (* *p*  <  0.05; ** *p*  <  0.01; *** *p*  <  0.001; **** *p*  <  0.0001).

**Figure 6 vaccines-13-00780-f006:**
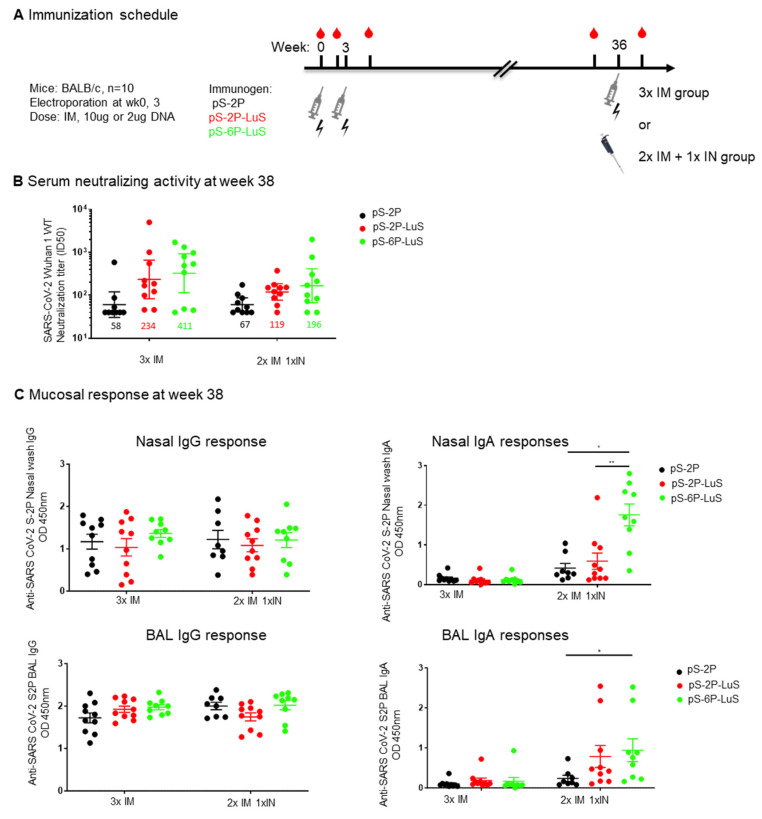
IM DNA Immunization and protein IN boost elicit systemic and mucosal immune response. (**A**) Extended schedule for the study shown in Figure 5. The 10 µg DNA and the 2 µg DNA groups were further boosted at Week 36 with a third IM DNA or 2 μg IN protein boost, respectively. (**B**) The serum-neutralizing activity against pseudotyped SARS-CoV-2 Wuhan virus at Week 38 is shown. (**C**) Mucosal IgG and IgA response at Week 38 as measured with ELISA. IgG response and IgA response were measured at 1:3 dilution of nasal wash or at 1:6 dilution of BAL. Geometric mean and 95% CI are shown for neutralizing activity, mean and SEM are shown in C. Data were analyzed by Kruskal–Wallis test followed by Dunn’s multiple comparison test. (* *p*  <  0.05; ** *p*  <  0.01).

**Table 1 vaccines-13-00780-t001:** Immunogen construct and protein yield information.

Construct	Arrangement and Sequence of SARS-CoV-2, Linker and LuS	Yield Post Lectin (mg/L)
S-2P-LuS01	S-2P (1-1147)-3AA linker-LuS	0.77
S-2P-LuS02	S-2P (1-1147)-7AA linker-LuS	1.18
NTD-LuS	NTD (1-308)-14AA linker-LuS	0.42
S-6P-LuS13	S-6P (1-1206)-7AA linker-LuS	3.44
S-6P-LuS14	S-6P (1-1206)-15AA linker-LuS	2.03

## Data Availability

The materials and resources used in this study are available and can be provided upon request from the corresponding authors.

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
