# Peer review of "Systemic and Mucosal Humoral Immune Responses to Lumazine Synthase 60-mer Nanoparticle SARS-CoV-2 Vaccines"

_vaccines, 2025, doi:10.3390/vaccines13080780_

Round 1
Reviewer 1 Report
Comments and Suggestions for Authors
These are my comments, most minor and two major.
Line 30-32, "These spike-LuS nanoparticles elicited significantly higher SARS-CoV-2 neutralizing activity than two intramuscular doses of purified spike trimers in vaccinated mice", I assume the former was a mucosal immunization, if so, indicate this.
Line 119, "1L", separate units from dimensions (1 L), do this throughout the document
Line 125, "80ml", use 80 mL
Line 156, "pH7.4", use pH 7.4
Line 161, "1X PBS", use a times symbol instead of an X (1×), do this throughout the document
Line 216, use either ACE-2 or ACE2, not both
Line 228, "57,000x", use a times symbol again
Line 268, "elusion", elution?
Line 269, "Galanthus nivalis", use italics
Figure 2, add standard deviations to all data, use mL instead of ml for the x-axis
Results, please include data regarding the nanovaccine stability. For this, measure size, polydispersity index, and zeta potential (using a zetasizer) of the formulation (at least the best one) at different time points at 4, 25 and 37 C.
Author Response
These are my comments, most minor and two major.
Response: We thank the reviewer for the supportive comments on our manuscript.
Line 30-32, "These spike-LuS nanoparticles elicited significantly higher SARS-CoV-2 neutralizing activity than two intramuscular doses of purified spike trimers in vaccinated mice", I assume the former was a mucosal immunization, if so, indicate this.
Response: We have revised the sentence to clarify that both were IM immunizations: After two intramuscular doses in vaccinated mice, these purified spike-LuS nanoparticles elicited significantly higher SARS-CoV-2 neutralizing activity than spike trimers.
Line 119, "1L", separate units from dimensions (1 L), do this throughout the document
Response: Revised on Line 126 and Line 131.
Line 125, "80ml", use 80 mL
Response: Revised on Line 132.
Line 156, "pH7.4", use pH 7.4
Response: Revised on Line 164.
Line 161, "1X PBS", use a times symbol instead of an X (1×), do this throughout the document
Response: This has been revised in seven places throughout the document.
Line 216, use either ACE-2 or ACE2, not both
Response: Revised on Line 228; now written as ACE2.
Line 228, "57,000x", use a times symbol again
Response: Revised on Line 240.
Line 268, "elusion", elution?
Response: Thank you for noting the typo. This has been changed to elution on Line 284.
Line 269, "Galanthus nivalis", use italics
Response: Revised on Line 285.
Figure 2, add standard deviations to all data, use mL instead of ml for the x-axis
Response: We have changed the x-axis labelling to mL. The data depicted in Figure 2 shows seven point serial dilutions of antibodies for each antigen. While this experiment was repeated only once in this manner, it was repeated in the context of the stability included in the new Supplement Figure 2 (see details in the next response)
Results, please include data regarding the nanovaccine stability. For this, measure size, polydispersity index, and zeta potential (using a zetasizer) of the formulation (at least the best one) at different time points at 4, 25 and 37 C.
Response: We appreciate the reviewer’s interest in the stability of the nanoparticle vaccine. We added the stability data at different temperatures, measured as a function of the nanoparticle’s ability to bind ACE2 and antibody, in a new Supplemental Figure 2. We modified the method section of the paper and included a description of the data as follows:
Methods section, Line 167-169:
“For stability analysis, S-2P or nanoparticles were diluted to 4 µg/mL in PBS and incubated at different temperatures from 4 °C to 65 °C for 10 minutes. Their recognition by ACE2 and monoclonal antibody of LY-CoV555 post the heat treatment was measured with ELISA.”
Main text, Line 314-321
“To test the thermostability of the vaccines, S-2P soluble trimers, S-2P-LuS02, and S-6P-LuS13 nanoparticles were incubated at temperatures ranging from 4 °C to 65 °C. All three vaccines exhibited strong binding to ACE2 after 10 minutes of exposure to 4 °C, 45 °C, and 50 °C; heating to 65 °C significantly reduced the binding. The S-6P-LuS13 nanoparticles maintained relatively better binding to ACE2 than S-2P and S-2P-LuS02, particularly after exposure to elevated temperatures of 45 and 50 °C. All three vaccines efficiently bound the LY-555 antibody after incubation at all temperatures tested, even at 65 °C.”
Supplementary Figure 2. Stability of S-2P soluble trimer and nanoparticle vaccines when heated. Purified immunogens were heated at temperatures ranging from 4 °C to 65 °C for 10 minutes and then coated on ELISA plates to measure their binding to ACE2-hFc or anti-COVID neutralizing antibody LY-CoV555.
Reviewer 2 Report
Comments and Suggestions for Authors
The manuscript "Systemic and Mucosal Humoral Immune Responses to Lumazine Synthase 60-mer Nanoparticle SARS-CoV-2 Vaccines" by Cheng Cheng et al., studies the role of LuS nanoparticles as potential SARS-CoV-2 vaccines. This study is important for vaccine development as it addresses the critical need for vaccines capable of eliciting both systemic and mucosal immunity to prevent SARS-CoV-2 infection and transmission. By leveraging the multivalent nanoparticle vaccine platform, which has proven successful in highly protective hepatitis B and human papillomavirus vaccines, the authors demonstrate the potential of spike-LuS nanoparticles to elicit robust neutralizing activity and stimulate mucosal IgA responses. The innovative self-assembling design and superior immunogenicity observed in this study highlight the promise of LuS nanoparticles as a transformative strategy for combating SARS-CoV-2 and its emerging variants, demonstrating the way for next-generation vaccine platforms. However, I have several reservations regarding the findings of this study.
Here are my major concerns:
1. What is the stability of this nanoparticle vaccine? How stable is it under room temperature and at 4°C?
2. Did the authors evaluate the neutralizing capacity of the mucosal samples?
3. Did the authors assess the protective efficacy through an animal challenge experiment? Additionally, does the administration of this nanoparticle vaccine cause any pathological side effects?
Author Response
Comments and Suggestions for Authors
The manuscript "Systemic and Mucosal Humoral Immune Responses to Lumazine Synthase 60-mer Nanoparticle SARS-CoV-2 Vaccines" by Cheng Cheng et al., studies the role of LuS nanoparticles as potential SARS-CoV-2 vaccines. This study is important for vaccine development as it addresses the critical need for vaccines capable of eliciting both systemic and mucosal immunity to prevent SARS-CoV-2 infection and transmission. By leveraging the multivalent nanoparticle vaccine platform, which has proven successful in highly protective hepatitis B and human papillomavirus vaccines, the authors demonstrate the potential of spike-LuS nanoparticles to elicit robust neutralizing activity and stimulate mucosal IgA responses. The innovative self-assembling design and superior immunogenicity observed in this study highlight the promise of LuS nanoparticles as a transformative strategy for combating SARS-CoV-2 and its emerging variants, demonstrating the way for next-generation vaccine platforms. However, I have several reservations regarding the findings of this study.
Here are my major concerns:
1.What is the stability of this nanoparticle vaccine? How stable is it under room temperature and at 4°C?
Response: We appreciate the reviewer’s interest in the stability of the nanoparticle vaccine. We added the stability data at different temperatures, measured as a function of the nanoparticle’s ability to bind ACE2 and antibody, in a new Supplemental Figure 2. We modified the method section of the paper and included a description of the data as follows:
Methods section, Line 167-170:
“For stability analysis, S-2P or nanoparticles were diluted to 4 µg/mL in PBS and incubated at different temperatures from 4 °C to 65 °C for 10 minutes. Their recognition by ACE2 and monoclonal antibody of LY-CoV555 post the heat treatment was measured with ELISA.”
Main text, Line 314-321
“To test the thermostability of the vaccines, S-2P soluble trimers, S-2P-LuS02, and S-6P-LuS13 nanoparticles were incubated at temperatures ranging from 4 °C to 65 °C. All three vaccines exhibited strong binding to ACE2 after 10 minutes of exposure to 4 °C, 45 °C, and 50 °C; heating to 65 °C significantly reduced the binding. The S-6P-LuS13 nanoparticles maintained relatively better binding to ACE2 than S-2P and S-2P-LuS02, particularly after exposure to elevated temperatures of 45 and 50 °C. All three vaccines efficiently bound the LY-555 antibody after incubation at all temperatures tested, even at 65 °C.”
Supplementary Figure 2. Stability of S-2P soluble trimer and nanoparticle vaccines when heated. Purified immunogens were heated at temperatures ranging from 4 °C to 65 °C for 10 minutes and then coated on ELISA plates to measure their binding to ACE2-hFc or anti-COVID neutralizing antibody LY-CoV555.
- Did the authors evaluate the neutralizing capacity of the mucosal samples?
Response: We have not measured the neutralizing activity of the mucosal samples due to the interference of the mucosal fluid in the neutralization assay. We added this information in the discussion section from Line 548-552:
“The neutralizing activity of the mucosal samples was not determined in this study due to interference of the mucosal fluid in the neutralization assay. Future studies will explore optimal nanoparticle regimens to prevent SARS-CoV-2 infection in animal models and the neutralizing activity in the mucosal compartment.”
- Did the authors assess the protective efficacy through an animal challenge experiment? Additionally, does the administration of this nanoparticle vaccine cause any pathological side effects?
Response: We have not measured the protective efficacy in an animal challenge experiment. However, it has been shown in mice that a serum ID50 titer of 819 protects mice from SARS-CoV-2 challenge (Cober et al., Nature (2020)). The ID50 titer achieved in this study is within this protective threshold. No phthological side effects were observed in the study animals. We have added a discussion of the neutralizing antibody correlates of protection in this model to the manuscript in Lines 554-557: “Compared to the reported ID50 neutralizing titer achieved with mRNA vaccine which is protective in mouse (819, geometric mean), the S-6P-LuS14 nanoparticles elicited similar range of high titers (Figure 3D with ID50 geometric means from 487 to 1,134 depending on dosage).”
Reviewer 3 Report
Comments and Suggestions for Authors
The authors present a study evaluating the immunogenicity of lumazine synthase (LuS) 60-mer nanoparticle vaccines displaying the SARS-CoV-2 spike protein. They report that these nanoparticles elicit robust systemic and mucosal immune responses in mice, demonstrating potential as a vaccine platform against COVID-19. Please find some of my comments below.
1) Please clarify how this vaccine platform compares to currently approved mRNA vaccines in terms of immunogenicity and scalability.
2) Elaborate on the rationale for choosing the lumazine synthase (LuS) 60-mer as the nanoparticle scaffold, and whether alternative scaffolds were considered.
3) Consider expanding on the implications of the observed mucosal IgA responses for real-world protection against SARS-CoV-2 infection.
4) Include a discussion on the potential cross-reactivity or protection against emerging SARS-CoV-2 variants beyond B.1.351.
5) Detail any safety considerations of using the LuS nanoparticle platform, especially regarding pre-existing immunity or potential autoimmunity.
6) Provide more information on the quantitative data supporting the claimed superiority of the LuS nanoparticle vaccines over spike trimers.
7) Discuss how the vaccine’s stability and production yield compare to conventional vaccine platforms to support its feasibility.
8) Comment on the limited durability of the neutralizing antibody responses, particularly after the second boost, and discuss strategies to enhance longevity.
Author Response
The authors present a study evaluating the immunogenicity of lumazine synthase (LuS) 60-mer nanoparticle vaccines displaying the SARS-CoV-2 spike protein. They report that these nanoparticles elicit robust systemic and mucosal immune responses in mice, demonstrating potential as a vaccine platform against COVID-19. Please find some of my comments below.
1) Please clarify how this vaccine platform compares to currently approved mRNA vaccines in terms of immunogenicity and scalability.
Response: We have added a discussion of this point to the manuscript Lines 554 to 560:
“Compared to the reported ID50 neutralizing titer achieved with mRNA vaccine in mice (819, geometric mean), the S-6P-LuS14 nanoparticles elicited a similar range of high titers (ID50 geometric means from 487 to 1,134, depending on vaccine dose; Figure 3D). Although we used transient transfection to produce these nanoparticle vaccines with a vaccine yield of above 1 mg/L, the single-component self-assembling characteristics of this platform readily support the creation of a stable producer cell line to support large-scale production.”
2) Elaborate on the rationale for choosing the lumazine synthase (LuS) 60-mer as the nanoparticle scaffold, and whether alternative scaffolds were considered.
Response: The information is added in a revised Lines 82 to 87:
“The main rationales for choosing LuS 60-mer as the nanoparticle platform in our study are: 1) the bacterial enzyme has no human analogues with low risk for auto-immune reactions, 2) the 60-mer nanoparticles have already been safely tested in human phase I HIV-1 vaccine trials without severe adverse effect and 3) the 60-mer nanoparticle used here is thermostable which is advantageous for manufacture and storage compared to other platforms.”
3) Consider expanding on the implications of the observed mucosal IgA responses for real-world protection against SARS-CoV-2 infection.
Response: In our study, we demonstrate that S-6P nanoparticle boosting regimens elicited significant IgA titers in the nasal cavity and BAL in the absence of a mucosal adjuvant. We have revised the Discussion Section to include the requested analysis.
Lines 548 to 554:
“The neutralizing activity of the mucosal samples was not determined in this study due to interference of the mucosal fluid in the neutralization assay. Future studies will explore optimal nanoparticle regimens to prevent SARS-CoV-2 infection in animal models and the neutralizing activity in the mucosal compartment. Furthermore, the exact level of IgA needed for protection and the durability of the IgA response are not known. A proper mucosal adjuvant should further boost such responses both for potency and longevity for protective effect. ”
4) Include a discussion on the potential cross-reactivity or protection against emerging SARS-CoV-2 variants beyond B.1.351.
Response: The information is added in the Discussion section, Lines 512 to 516.
”A third boost with a B.1.351 variant nanoparticle vaccine significantly enhanced ELISA IgG titer and neutralizing activity to the Wuhan strain and multiple variants (Alpha, Beta, Delta), suggesting that heterologous cross-variant protection is feasible. The nanoparticle vaccines can also be updated to include contemporary S protein sequences.”
5) Detail any safety considerations of using the LuS nanoparticle platform, especially regarding pre-existing immunity or potential autoimmunity.
Response: The information is added in the discussion section, Line 488-490.
“The self-assembling eOD-GT8 60mer has been GMP manufactured and shown to be safe, tolerable, and immunogenic in eliciting VRC01 precursor B-cells in a recent Phase-1 trial [50]. Anti-LuS immune responses were detected in eOD-GT8 60mer vac-cinated participants without adverse effects [66]. As hyperthermophilic bacterium Aquifix aeolicus is not a member of the human microbiome, pre-existing immunity to LuS 60mer in humans and the risk of auto reactivity is low. Thus, the single-component LuS genetic fusion platform is a technology with the potential to be widely applicable to vaccines for other emerging and re-emerging pathogens.”
6) Provide more information on the quantitative data supporting the claimed superiority of the LuS nanoparticle vaccines over spike trimers.
Response: The information is added in the discussion section, Lines 346 to 351.
“At the lowest vaccine dose (0.08 μg), nanoparticle immunogens elicited significantly higher neutralizing activity (geometric mean ID80 titer of 305 and 204 for the S-2P nanoparticles and S-6P nanoparticles, respectively) than the recombinant trimer, which did not elicit detectable neutralizing activity (<40, Figure 3D, p<0.01). The nanoparticles are at least 5-fold more potent than S-2P trimers as measured with ID80 titers.”
7) Discuss how the vaccine’s stability and production yield compare to conventional vaccine platforms to support its feasibility.
Response: We have added a discussion of this point to the manuscript Lines 554 to 560:
“Compared to the reported ID50 neutralizing titer achieved with mRNA vaccine in mice (819, geometric mean), the S-6P-LuS14 nanoparticles elicited a similar range of high titers (ID50 geometric means from 487 to 1,134, depending on vaccine dose; Figure 3D). Although we used transient transfection to produce these nanoparticle vaccines with a vaccine yield of above 1 mg/L, the single-component self-assembling characteristics of this platform readily support the creation of a stable producer cell line to support large-scale production.”
8) Comment on the limited durability of the neutralizing antibody responses, particularly after the second boost, and discuss strategies to enhance longevity.
Response: The information is added in the discussion section of the revised manuscript in Lines 503 to 509.
“As shown in Figure 4, serum neutralizing activity peaked at week 5 after the second immunization and decreased by 10-fold by week 31. We did not collect data from a sufficient number of time points to estimate the durability of the response with precision. This limited durability of the response may be animal model-specific or modulated using an optimal adjuvant. Further studies are required to measure and potentially extend the durability of the S protein antibody response.”
Reviewer 4 Report
Comments and Suggestions for Authors
The authors present a comprehensive study of the efficacy in animal studies of nanoparticle vaccines for Covid based on the assembly of the Lumazine Synthase into 60-mers. Genetic manipulation is used to create fusion proteins with S protein with 2 or 6 stabilizing prolines. As explained by the authors, the LuS system is not found in humans and could become a platform technology for vaccines for other and emerging pathogens. The data are presented in very informative graphs and charts that are nicely color coded. The results of good immunogenicity in mice suggest that that the study has translational significance. A few items could be discussed to refine the mansuscript:
(1) Concerning the presentation of S protein versus NTD on the nanoparticle surface at 3-fold vertices, discuss the coverage and packing in terms of how much surface of LuS is exposed in each case. This is of interest since S protein is much larger than NTD.
(2) Which sugars lead to the choice of the GNA lectin?
(3) How does this vaccine compare to the few others reported based on proteins assembled into nanoparticles such as those based on tobacco virus and murine leukemia virus?
Author Response
Comments and Suggestions for Authors
The authors present a comprehensive study of the efficacy in animal studies of nanoparticle vaccines for Covid based on the assembly of the Lumazine Synthase into 60-mers. Genetic manipulation is used to create fusion proteins with S protein with 2 or 6 stabilizing prolines. As explained by the authors, the LuS system is not found in humans and could become a platform technology for vaccines for other and emerging pathogens. The data are presented in very informative graphs and charts that are nicely color coded. The results of good immunogenicity in mice suggest that that the study has translational significance. A few items could be discussed to refine the mansuscript:
Response: We thank the reviewer for the nice summary of the manuscript.
Concerning the presentation of S protein versus NTD on the nanoparticle surface at 3-fold vertices, discuss the coverage and packing in terms of how much surface of LuS is exposed in each case. This is of interest since S protein is much larger than NTD.
Response: The information is added in the discussion section, Line 521 to 528.
On the other hand, based on molecular modelling, it is estimated that 31% of the S-2P-LuS nanoparticle surface is covered with the spike, while for NTD-LuS nanoparticles surface is 35% covered by the NTD. Although the S protein trimer is considerably larger than the NTD protein, the actual LuS surface area exposed in the S protein nanoparticle verses the NTD nanoparticle is quite similar. This is in part because the S protein monomers are held tightly together in their respective trimers, particularly at the trimer base and also because of the flexibility provided by the glycine-rich linkers.
Which sugars lead to the choice of the GNA lectin?
Response: The information is added in the Main text of the revised manuscript, Lines 278 to 281:
As there are 22 potential N-linked and two potential O-linked glycosylation sites per spike monomer, the spike glycoprotein contains many α-1,3 mannose residues which can be bound by Galanthus nivalis lectin.
How does this vaccine compare to the few others reported based on proteins assembled into nanoparticles such as those based on tobacco virus and murine leukemia virus?
Response: The information is added in the Discussion section of the revised manuscript, Lines 509 to 512.
“The immune responses elicited by our nanoparticle vaccines are similar to the responses by other viral like particles, including the Tobacco Mosaic virus-like nanoparticle (Royal et al Vaccines 2021, 9(11), 1347; https://doi.org/10.3390/vaccines9111347) and the murine leukemia virus (MLV)-based enveloped virus-like particles (eVLPs) (Fluckiger et al Vaccine, 39(35):4988-5001. doi: 10.1016/j.vaccine.2021.07.034. Epub 2021 Jul 16.)
Round 2
Reviewer 1 Report
Comments and Suggestions for Authors
The authors have addressed all my concerns.
Reviewer 2 Report
Comments and Suggestions for Authors
I have reviewed the authors' responses to my comments and the revisions made to the manuscript. I am satisfied with their responses and the changes provided in the revised manuscript. All my concerns have been adequately addressed.
Reviewer 3 Report
Comments and Suggestions for Authors
The authors have addressed my queries.